# One-Step Green Synthesis of Water-Soluble Fluorescent Carbon Dots and Its Application in the Detection of Cu^2+^

**DOI:** 10.3390/nano12060958

**Published:** 2022-03-14

**Authors:** Saheed O. Sanni, Theo H. G. Moundzounga, Ekemena O. Oseghe, Nils H. Haneklaus, Elvera L. Viljoen, Hendrik G. Brink

**Affiliations:** 1Department of Chemical Engineering, Faculty of Engineering, Built Environment and Information Technology, University of Pretoria, Pretoria 0028, South Africa; 2Biosorption and Wastewater Treatment Research Laboratory, Department of Chemistry, Faculty of Applied and Computer Sciences, Vaal University of Technology, Vanderbijlpark 1900, South Africa; theo.hernan6@gmail.com (T.H.G.M.); elverav@vut.ac.za (E.L.V.); 3Institute for Nanotechnology and Water Sustainability, College of Science, Engineering and Technology, Florida Campus, University of South Africa, Johannesburg 1709, South Africa; eoseghe@gmail.com; 4Institute of Chemical Technology, Freiberg University of Mining and Technology, Leipziger Straße 29, 09599 Freiberg, Germany; nils-hendrik.haneklaus@extern.tu-freiberg.de; 5Td Lab Sustainable Mineral Resources, University for Continuing Education Krems, Dr.-Karl-Dorrek-Straße 30, 3500 Krems an der Donau, Austria

**Keywords:** renewable biowaste, carbon dots, microwave pyrolysis, Cu^2+^ ion detection

## Abstract

Renewable biowaste-derived carbon dots have garnered immense interest owing to their exceptional optical, fluorescence, chemical, and environmentally friendly attributes, which have been exploited for the detection of metals, non-metals, and organics in the environment. In the present study, water-soluble fluorescent carbon dots (CDs) were synthesized via facile green microwave pyrolysis of pine-cone biomass as precursors, without any chemical additives. The synthesized fluorescent pine-cone carbon dots (PC-CDs) were spherical in shape with a bimodal particle-size distribution (average diameters of 15.2 nm and 42.1 nm) and a broad absorption band of between 280 and 350 nm, attributed to a π-π* and n-π* transition. The synthesized PC-CDs exhibited the highest fluorescent (FL) intensity at an excitation wavelength of 360 nm, with maximum emission of 430 nm. The synthesized PC-CDs were an excellent fluorescent probe for the selective detection of Cu^2+^ in aqueous solution, amidst the presence of other metal ions. The FL intensity of PC-CDs was exceptionally quenched in the presence of Cu^2+^ ions, with a low detection limit of 0.005 μg/mL; this was largely ascribed to Cu^2+^ ion binding interactions with the enriched surface functional groups on the PC-CDs. As-synthesized PC-CDs are an excellent, cost effective, and sensitive probe for detecting and monitoring Cu^2+^ metal ions in wastewater.

## 1. Introduction

Elevated levels of heavy metals in surface and ground water have been reported by many researchers, and are attributed to their increased presence in industrial effluents, increased mining activities, and spontaneous natural and man-made activities [1,2]. Heavy metals are non-biodegradable and accumulate readily in the hydro- and bio-sphere, thus contributing to a wide variety of side effects including ecological consequences and diseases [3,4]. One heavy metal of interest is Copper (Cu^2+^), which is known to accumulate in fish (especially the gills, causing mortality through respiratory disruption) [3,4,5], bacteria, and viruses [6]. Gastrointestinal disturbance; liver and kidney damage; and various neurodegenerative diseases such as Alzheimer’s, Wilson’s, and Parkinson’s diseases have been attributed to long term exposure to Cu^2+^ in humans [7,8]. Based on the challenges highlighted above the U.S. Environmental protection Agency (EPA) and the World Health Organisation (WHO) have set the limit of copper in drinking water to 1.3 µg/mL (20 µM) [9] and 2.0 µg/mL (32 µM), respectively [10]. The permitted Cu^2+^ level in normal human blood is typically between 1–1.5 µg/mL [11]. For these reasons, it is necessary to develop an efficient, highly sensitive, and selective method for the detection very low concentrations of Cu^2+^ in the environment.

Lately, a host of analytical approaches have been utilized for assaying Cu^2+^ metal ions in aqueous solution comprising electrochemical [12], atomic fluorescence spectrometry [13], mass spectrometry [14], and atomic absorption spectroscopy [15] methods. Nevertheless, these methods are time consuming and complicated, and require expensive equipment for the routine determination of Cu^2+^. Fluorescence spectroscopy has been shown to be an effective detection technique that could meet most of the requirements for heavy metal detection of Cu^2+^ ions, which are of ecological concern. This technique has the distinct advantage of intrinsic sensitivity, a wide linear dynamic range, easy operation, selectivity, and an intensive capacity for rapid real-time monitoring [11,16,17]. The fluorescent assay approach utilizing nanosensor-materials, such as the conjugated derivatives of rhodamine B and semicarbazide [18], and semiconductor quantum dots (SQDs) [19,20], have been applied for the detection of Cu^2+^ ions in aqueous solution. Unfortunately, these aforementioned probes have the disadvantage of complicated synthesis procedures and photobleaching [21]. SQDs have other major disadvantages including potential toxicity, chemical instability, intrinsic blinking [20], and insolubility in aqueous solution [22].

Carbon dots (CDs), a family of fluorescent carbon nanomaterials, have generated wide interest due to their advantages of excellent water solubility, good photo-stability, high chemical inertness, low cytotoxicity, and biocompatibility [23,24,25]. However, conservation of the environment has prompted the focus of CD synthesis to adopt greener approaches via the utilization of renewable biowaste precursors, instead of the usage of chemicals [22,26,27]. CD-synthesis from these renewable biowaste materials also comes with added benefits, including reducing the cost of waste disposal, and the recycling of these wastes for environmental protection, thus promoting large-scale and low-cost production of CDs [23,28,29]. CDs have been synthesized from the pyrolysis of sago waste [24], chemical treatment of prawn shells [7], hydrothermal treatment of vegetables [23], and hydrothermal treatment of fresh bamboo leaves [30] for the detection of Cu^2+^ ions in aqueous solution. However, most of these methods for CD production have had drawbacks, including the utilization of large quantities of strong acids [31,32]; complex preparation methods involving multiple steps [32,33]; and expensive surface functionalization steps to improve their fluorescent properties and Cu^2+^ sensitivity [31,34]. The application of these aforementioned corrosive agents and multistep procedures, thus, raises environmental and economic concerns, thereby limiting their practical applications. It is therefore desirable to produce CDs using a chemical-free, simple, one-step, rapid, and low-energy consuming method that eliminates additional modifications, to improve the fluorescence properties of CD sensitivity and selectivity for Cu^2+^ ion detection. Microwave pyrolysis is considered a promising heating approach, which is primarily ascribed to its minimal expense, simplicity, nontoxicity, harmlessness to the ecosystem, and capacity to create luminescent CDs with high quantum yield [34]. Lots of investigations on integrated CDs from biomass materials with high luminescent properties, and quantum yield obtained through micro-wave pyrolysis applied in a fluorescence sensor, and cell imaging, have been reported [35,36].

Pine-cone biowaste is naturally abundant within the South African environment; is plentiful in cellulose, hemicellulose and lignin components; and was exploited as a carbon source for various applications in [37,38,39]. Appropriately, the researchers expected that this biowaste material would be appropriate for the synthesis of fluorescent CDs in the selective and sensitive detection of Cu^2+^ ions.

The current study focuses on the synthesis of water-soluble highly fluorescent pine-cone carbon dots (PC-CDs), using a one-step green microwave pyrolysis method without any chemical additives or passivating agents in the preparation approach (Figure 1). The synthesized PC-CDs were subsequently evaluated for selective detection of Cu^2+^ ions in synthetic aqueous solutions. In addition, the PC-CDs were used to quantify the concentration of Cu^2+^ metal in a representative authentic wastewater sample.

## 2. Experimental Procedure

### 2.1. Materials

Agricultural biomass pine-cone (PC) was collected from the parking space of the Vaal University of Technology, South Africa. Ultrapure water was utilized throughout the performed experiments. Rhodamine B with 99% purity was purchased from Sigma-Aldrich (St. Louis, MO, USA). Nitrate salts of various metal cations comprising Ba^2+^, Cd^2+^, Co^2+^, Cu^2+^, Cr^3+^, Cs^+^, Fe^2+^, Fe^3+^, Hg^2+^, K^+^, Mg^2+^, Mn^2+^, Na^+^, Ni^2+^, Pd^2+^ and Zn^2+^ with 99% purity were bought from Sigma-Aldrich.

### 2.2. Synthesis of PC-CDs

The PCs were washed several times to remove impurities from the material and dried in an oven at 90 °C for 48 h. The scales on the cones were removed and crushed using a pulverizer to make a fine powder. The PC powder was then sieved to less than 300 µm and used in CDs synthesis. The sieved pine-cone (10 g) was placed in a 250 mL Duran quartz bottle and microwave pyrolysis (LG MH8042GM, maximum power of 1000 W at a frequency of 2450 MHz, Seoul, Korea) was performed at 1000 W for 1 h under an inert atmosphere by purging with nitrogen gas. After the pyrolyzed material was cooled to room temperature, 3 g of pyrolyzed sample was transferred into 150 mL ultrapure water, sonicated for 30 min in a Branson 2800 sonication water bath (Emerson, St. Louis, MO, USA), and centrifuged at 10,000 rpm for 60 min to dispose of the supernatant and bigger carbon residues. The supernatant was purified using a 0.22 µm filter membrane, then freeze-dried inside a SCANVAC CoolSafe™ (Scientific Labs, Nottingham, UK) freeze dryer to obtain a water-soluble solid extract of PC-CDs. The quantum yield measurement of the PC-CDs is described in Appendix A.

### 2.3. Characterization of PC-CDs

The size and morphology of the PC-CDs were obtained using a Tecnai 20 transmission electron microscope (TEM) (FEI Company, Hillsboro, OR, USA) operating at an acceleration voltage of 200 kV. The particle size distribution was obtained by image analysis of the obtained TEM image using the ImageJ software package (National Institutes of Health, Bethesda, MD, USA). X-ray diffraction (XRD) phases for PC-CDs were obtained using a Bruker diffractometer AXS (Billerica, MA, USA) with CuKα radiation source. The PC-CDs’ functional groups were investigated using a Fourier transform infrared (FTIR) spectrometer (Perkin Elmer Spectrum 400, Waltham, MA, USA) within a range of 4000–400 cm^−1^. Thermogravimetric analysis (TGA) and differential scanning calorimetric (DSC) measurements were conducted to examine the thermal behavior of PC-CDs using a PerkinElmer STA 6000 thermal analyzer (Waltham, MA, USA). Raman spectra of the PC-CDs were determined using a Horiba LabRAM HR Evolution (Kyoto, Japan). The ultraviolet –visible absorption spectra of the PC-CDs were measured by using double-beam UV–visible spectrophotometry between the wavelengths of 200 and 800 nm. The photoluminescence (PL) intensity spectra of PC-CDs were measured by using a fluorescence spectrophotometer (FP-8600 Spectrofluorometer, Jasco, Midrand, South Africa). The Zeta potential of the PC-CDs was analyzed using a Malvern Zetasizer Nano (Malvern, UK).

### 2.4. Fluorescence Sensing of Cu^2+^

The sensing application of the PC-CD probe was run on Cu^2+^ detection at room temperature in phosphate buffer solution (PBS; pH 4). For one run, 750 µL of the PC-CD (2 mg/mL) was added to 500 µL of PBS inside a centrifuge tube, followed by the addition of 20 µL of a different concentration of Cu^2+^. Fluorescence emission intensity was measured after 10 min of incubation at room temperature. For interference experiments, different metal ions with concentration of 25 µg/mL were added to 750 µL of the PC-CD solution and mixed with Cu^2+^ for fluorescence measurement. The fluorescence (FL) spectra of the solution dispersion were recorded at an excitation wavelength of 360 nm between a range of 300–700 nm.

To evaluate the potential of the proposed approach for the analysis of Cu^2+^ in a real wastewater system, the PC-CD nanosensors were applied for the detection of Cu^2+^ in a representative wastewater sample. The real wastewater sample was purified by means of filtration using a 0.22 µm membrane (Millipore syringe filters Millipore Sigma, Burlington, MA, USA) prior to the detection of Cu^2+^. Different Cu^2+^ concentrations were spiked in the above filtered water sample and the fluorescence measurements were carried out in triplicate at room temperature.

## 3. Results and Discussion

### 3.1. Characterization of PC-CDs

PC-CDs were synthesized using green microwave pyrolysis of PC biowaste with a high carbon content of 44.1% [40], without the addition of chemical reagents or passivating agents, as presented in Figure 1. The morphology and particle size distribution of the synthesized PC-CDs were assessed by TEM analysis, as depicted in Figure 1A. As shown in Figure 1A, the PC-CDs are spherical in shape and well dispersed. The synthesized PC-CDs’ particle sizes had a bimodal distribution (Figure 1B), with the smaller sized population (circled red) having particles smaller than 25 nm and an average diameter *d*_50_ = 15.2 ± 2.7 nm (average ± standard deviation) (Gaussian model fit with R^2^ = 0.9984). The smaller particles constituted most of the particles in the sample (81.45% of the total particles) and were distributed in a very narrow range around the *d*_50_, as demonstrated by the relatively small standard deviation. The larger-sized population, an amorphous structure with a large spherical shape (Figure 1A), constituted the remainder of the sample (18.55%), and had particles much more evenly spread between 25 and 51 nm (circled green). The larger particle population had a *d*_50_ = 42.1 ± 9.3 nm (Gaussian model fit with R^2^ = 0.9999). Similar observation is evident with other reported works utilizing biomass in CD preparation, whereby small byproducts of the reaction are still present in the supernatant solution after centrifugation [41,42].

The XRD pattern of PC-CDs (Figure 1C) shows a broad peak at 2θ = 22.7° and a discernable peak at 2θ = 43.6°. The peaks at 22.7° and 43.6° are assigned to the 002 and 101 lattice-planes. The 002 plane possess an interlayer spacing of ~0.37 nm and is larger than the bulk graphite with a spacing of ~0.33 nm. The assigned peaks evidence the presence of highly disordered carbon atoms and a graphitic structure in the CDs [36,43]. The crystalline size of PC-CDs was calculated using the Debye–Scherrer equation, as presented below (Equation (1)), and found to be 4.51 nm.
(1)$$D=\frac{K\lambda }{\beta Cos\theta }$$

Herein, *D* represents the crystallite size, *K* is assigned to the dimensionless shape factor whose factor is close to unity (0.94), *λ* corresponds to the X-ray wavelength used (1.5406 Å), and *β* is the full width at half maxima (FWHM) at 2θ.

FTIR spectroscopy was employed for the identification of the functional groups present in PC-CDs, and the spectra is presented in Figure 1D. The synthesized PC-CDs exhibited broad vibrational peaks at 3396 cm^−1^ and 3171 cm^−1^, attributed to the –OH/–NH, and the weak peak at 2900 cm^−1^ can be attributed to C–H stretching vibration [44]. The stretching vibrational band of –C=O and C=C bending vibrations of benzene rings were observed at 1695 cm^−1^ and 1575 cm^−1^, respectively [45,46]. The characteristic peak of –OCH_3_ appeared at 1423 cm^−1^ [47], while the vibrational peaks at 1030 and 814 cm^−1^ were indicative of C–O–C stretching [45] and C–H stretching vibrations [22]. The presence of enriched hydroxyl, carboxyl, and epoxy groups present in PC-CDs greatly assist in the stabilization and hydrophilicity of the PC-CDs in an aqueous solution [48]. The Zeta potential of the PC-CDs produced were negative at −10 mV as presented in Appendix A. The negative zeta potential value is ascribed to the abundance of negatively charged groups (hydroxyl, carboxyl, carbonyl, and epoxy) present on the surface of the PC-CDs. These results are consistent with CDs obtained from biomass materials such as green pepper seeds [49] and orange juice [28]. From these results, it can be posited that the negatively charged PC-CDs, with abundant functional groups, have the capacity to effectively bind Cu^2+^ metal ions in this study (Further discussions in Section 3.3).

The Raman spectra of the PC-CDs is presented in Figure 2A, revealing a D band around 1371 cm^−1^ and a strong G band at 1582 cm^−1^, attributed to the sp^3^ defects and sp^2^ hybridized carbon atoms in the PC-CDs material [50,51]. The intensity ratio of the two bands (I_D_/I_G_) was calculated to be 0.87 which highlights the high crystallinity of PC-CDs cleared of any carbonaceous impurities [29,52].

The TGA and DSC analysis of the PC-CDs is presented in Figure 2B. The PC-CDs exhibited two significant weight reduction steps. The small weight loss before 150 °C is attributed to the dissipation of water particles adsorbed on the PC-CDs. In the temperature interval between 150 and 400 °C, the weight loss was associated with the thermal decomposition of oxygen-containing functional groups on the surface of the PC-CDs [53]. The DSC degradation curve of the solid PC-CDs exhibited an intense exothermic peak at 265 °C, which is attributed to the decomposition of different functional groups present on the surface of the PC-CDs [54]. At higher temperatures (>265 °C) no further thermal degradation of the PC-CDs was observed up to the maximum temperature of 850 °C. This analysis further confirms the high thermal stability of PC-CDs synthesized from pine-cone biomass, which is in line with previous CDs prepared from biomass materials [54,55].

### 3.2. Optical Properties of Synthesized PC-CDs

Figure 3A shows the UV–Vis absorption spectra of PC-CDs. The UV–vis absorption spectra of the PC-CDs (Figure 3A) displayed an absorption band between 280 and 360 nm, which is coherent with other reported studies [47,56]. This corresponding absorption range is attributed to an n-π* transition of the C=O bonds and an π-π* transition of C=C bonds, confirming the formation of the PC-CDs [57,58,59]. The photoluminescence (PL) properties of the PC-CDs were explored based on the fact that CDs’ emission peaks are dependent on the excitation wavelength [48]. PC-CDs exhibited the strongest emission band at 430 nm when excited at 360 nm, which is attributed to the n-π* transition, as presented in Figure 3B. The FL properties of PC-CDs at different excitation wavelengths were explored from range of 300 nm to 405 nm with an increment of 15 nm. PC-CDs did not change considerably with an increased emission wavelength from 418 to 430 nm (as red shifting was lower compared with other studies [36,47]), while a decrease in FL intensity was observed upon increasing the excitation wavelength increase to 375–405 nm. These findings agree with results reported in the literature [60,61,62], as the excitation dependency properties of PC-CDs can be fine-tuned for multicolor imaging applications. The functional groups related to the PC-CDs might create a series of emissive traps among π and π* states of C=C on their surface that controls the emission of PC-CDs under illumination at a particular excitation wavelength [56]. Therefore, the fluorescence system of PC-CDs can be compelled by both the size effects and surface defects [63].

The quantum yield of the PC-CDs (as calculated in Appendix A) is estimated to be 17% using rhodamine B as the reference sample; this value is markedly better than other CDs prepared from different biomass materials, as presented in Table 1. This provides further evidence of the presence of aromatic hydrocarbons, the negatively charged oxygenated, and minute positively charged amino functional groups present on the surface of PC-CDs [64], which enhances the fluorescence attributes overall. The FL intensity for PC-CDs was observed at different pH values from 1–11 (via the addition of 0.1 M HCl and NaOH solution to the aqueous solution of the PC-CDs), as presented in Appendix A. An optimized FL intensity for PC-CDs was observed at pH 4, as a decreasing FL intensity trend was evident at a pH above 8 (strong base) and below 4 (strong acid). This is ascribed to the protonation, and deprotonation of the surface functional groups present on functional group of PC-CDs. The PC-CDs changed their functionality from positive to negative charges at a high pH, via dissociation of amino, and oxygenated functional groups [65,66], thus resulting in decreasing FL intensity. Consequently, pH 4 was chosen as the optimum pH for sensing studies in this work. The PC-CDs were placed under a UV-light irradiation of 365 nm for 240 min, with a 20 min difference interval. The PC-CD solution exhibited excellent photostability and avoided photobleaching as well, as the FL intensity was almost constant (Appendix A). The PC-CDs’ FL intensity continued to be stable under normal light storage up to 15 days, which also confirmed their resistance to quenching during storage (Appendix A), thus demonstrating their potential in sensing applications.

### 3.3. Fluorescence Response of PC-CDs to Cu^2+^

The fluorescence selectivity of the synthesized PC-CDs against a host of different metal cations was conducted by adding 25 µg/mL aqueous solution of Ba^2+^, Cd^2+^, Co^2+^, Cr^3+^, Cs^+^, Cu^2+^, Fe^2+^, Fe^3+^, Hg^2+^, K^+^, Mg^2+^, Mn^2+^, Na^+^, Ni^2+^, Pd^2+^, and Zn^2+^ to the synthesized PC-CD solution and subsequently measuring the FL intensities. The fluorescence properties of synthesized PC-CDs were not selective towards the majority of the metal cations (Figure 4A,B) except for Cu^2+^; the Cu^2+^ metal ion induced significant FL quenching of the PC-CDs. This exceptional affinity of the Cu^2+^ metal ions towards the functional groups (hydroxyl, carboxyl, and amino) present on the surface of PC-CDs, thus, pave way for chelated complex formation [73]. The developed PC-CD + Cu^2+^ complexes also promote a non-radiative electron-hole recombination break-down via an effective electron transfer that causes the fluorescence quenching of the PC-CDs [74,75].

The selectivity of Cu^2+^ in the presence of interfering metal ions was conducted through the addition of various cations, Cu^2+^, and PC-CDs, as presented in Appendix A. The presence of interfering metals on the PC-CDs (Appendix A) yielded no FL quenching of PC-CD material. However, Cu^2+^ metal showed an excellent FL-quenching effect on PC-CDs in the presence of interfering ions, and thus, demonstrated that PC-CD functional groups have a strong affinity towards binding Cu^2+^ ions in forming stable complexes. By considering the above-mentioned observations, it can be confirmed that the PC-CDs exhibited an exceptional high selectivity for Cu^2+^ over competing metal cation interference under their coexistent conditions.

### 3.4. Sensitivity of PC-CDs for Cu^2+^ Detection

The detection of the sensitivity of synthesized PC-CDs towards Cu^2+^ was conducted at different concentrations between a range of 0 to 25 µg/mL. There was a drastic reduction in the FL emission intensity of PC-CDs at 430 nm as the concentration of Cu^2+^ increased (Figure 5A), demonstrating excellent sensing of the metal ion in the system.

The fluorescence intensity is significantly reduced and the emissive wavelength of the fluorescent material along with the quencher do not change. Herein, the fluorescent intensity of synthesized PC-CDs is Cu^2+^ metal-ion-concentration-dependent. The fluorescence quenching of the PC-CDs by Cu^2+^ obeys a modified Stern−Volmer expression [76], while a non-linear exponential decay equation [77] was explored to fit the curve, as presented below (Equation (2)):(2)$$\frac{Fo}{F}=C+K_{1}\exp(K_{2}\left[S\right])$$

Where *Fo* and *F* are the fluorescence intensities of the PC-CDs at 430 nm in the absence and presence of Cu^2+^ ions, respectively. *K*_1_, *K*_2_, and *C* are dimensionless constants, and *S* is the concentration of Cu^2+^ (µg/mL). The concentration range of Cu^2+^ ions (2.5 to 22.5 µg/mL) fits Equation (1) (*Fo/F* = 1.493 + 0.009 exp (0.245 [*S*]) (Figure 5C) with a correlation coefficient of 0.9741. The limit of detection (LOD) was estimated to be 0.005 µg/mL at a signal-to-noise ratio of 3. Importantly, the obtained LOD is lower than the WHO recommendation for Cu^2+^ metal ions in drinking water (1.3 µg/mL) [76], which reveals that there is, indeed, potential for using PC-CDs prepared from PC biomass for assaying Cu^2+^ ions. The LOD value of PC-CD detection of Cu^2+^ metal ions is lower than, or comparable with, previously reported fluorescence assays for Cu^2+^ ion detection, as presented in Table 2.

### 3.5. Sensitivity of Cu^2+^ in Real Wastewater Effluent

To determine the sensitivity of the PC-CDs to Cu^2+^ in a real wastewater system, wastewater effluent from the university premises was used for a representative reference study and spiked with Cu^2+^ ions at different concentrations, from 0.5 to 6 µg/mL. It can clearly be seen that the fluorescence intensity of PC-CDs (Figure 6A) reduces as the concentration of Cu^2+^ increases from 0.5 to 6 µg/mL with a clear indication of good sensitivity towards Cu^2+^ ions. The Stern–Volmer plot (Figure 6B) fit Equation ($\mathrm{Fo}/F=1.475+0.081\left[{\mathrm{Cu}}^{\mathrm{2+}}\right]$) from 0.5 to 6 µg/mL with a correlation coefficient of 0.9965, with a LOD calculation of around 4.78 µg/mL. This is a remarkable correlation considering the host of viruses, bacteria, other metal ions, and organic compounds present in the wastewater sample. This result indicates that the as-synthesized PC-CDs can be successfully utilized as a potential sensing platform for monitoring Cu^2+^ ions in real wastewater effluent. The recovery percentage varies from 84.14 to 105.59%, respectively, as presented in Appendix A, which shows that this fluorescence sensor system has reasonable accuracy and an average relative standard deviation (RSD) value of 8.26%.

### 3.6. Quenching Mechanism

The fluorescence-quenching mechanism of normal CDs can be attributed to several factors such as inner filter effects, non-radiative recombination pathways, electron transfer, and ion-binding interaction [76]. The fluorescence-quenching mechanism of PC-CDs was assessed using the zeta potential and UV–visible spectroscopy, as presented in Appendix A. The addition of Cu^2+^ ions to the synthesized PC-CDs caused the zeta potential to change from −10 mV to 21 mV, which is ascribed to the complex formation [75]. The UV–Vis absorption spectra of PC-CDs were also measured by themselves and in the presence of 25 µg/mL copper ions. The broad absorption band of PC-CDs around 280 and 350 nm (Appendix A) showed definite reduction after adding the Cu^2^ metal ions, therefore indicating the formation of complexations between the functional groups of PC-CDs and Cu^2+^ ions. On the basis of this observation, the authors propose that the formation of complexation between PC-CDs and Cu^2+^ results in electron transfer between Cu^2+^ ions and PC-CDs (nonradiative energy), which plays a major role in quenching the FL of PC-CDs, as observed in Figure 5A [45]. This mechanism is represented in Figure 2.

The fluorescence-quenching mechanism of PC-CDs, thus, involves chemical interactions and the photophysical process, which are responsible for PC-CDs’ high selectivity and sensitivity towards Cu^2+^ ions in this study.

## 4. Conclusions

The green synthesis of water-soluble PC-CDs from the microwave pyrolysis of PC biowaste was successfully carried out in this study without the addition of corrosive reagents. The synthesized PC-CDs had a bimodal distribution with average sizes of 15.2 ± 2.7 nm (81.45% of particles) and 42.1 ± 9.3 nm (18.55% of particles), exhibited strong blue fluorescence under UV light illumination, and possessed a quantum yield of 17%. The synthesized PC-CDs possessed good solubility in water and photostability—which is ascribed to the surface functional groups—therefore making them a potential probe for sensor applications. In addition, PC-CDs exhibited excellent selectivity and sensitivity towards the detection of Cu^2+^ ions in aqueous solution, with a lower detection limit of 0.005 µg/mL—far lower than maximum acceptable level of Cu^2+^ in drinking water proposed by the WHO. The PC-CDs further exhibited excellent detection of Cu^2+^ ions in bona fide wastewater samples, thus highlighting their potential application for an environmental assay of this heavy metal ion.

## Data Availability

The data presented in this study are openly available in the University of Pretoria Research Data Repository at doi:10.25403/UPresearchdata.19345373.

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
