# Peer review of "One-Step Green Synthesis of Water-Soluble Fluorescent Carbon Dots and Its Application in the Detection of Cu2+"

_nanomaterials, 2022, doi:10.3390/nano12060958_

Round 1
Reviewer 1 Report
- Previous works on the microwave pyrolysis process for CD should be included in the introduction section. And the process should be fully explained.
- During the microwave pyrolysis process, what are the temperatures that the powder experience? The authors should further elaborate on the microwave pyrolysis process.
-
The quality (resolution) of TEM image is poor. These images should be replaced with higher-quality images. Furthermore, the TEM images do not support the claim that authors have made regarding the size of the CD. The average size 15 and 42 nm of these CD is not demonstrated. It is need to use Malvern sizer to check size in water solution directly.
-
It is possible to calculate average size of CD from the XRD data using Seliakov equanation. The authors should also include a histogram of the CD size range.
-
Experimental parameter such as particle size of the raw material, the density of the pellet, etc., plays an important role in the microwave pyrolysis process. The authors should explain why only a single experimental condition was used to synthesize the CD, and on what basis these experimental conditions were chosen?
- It is no need to demonstrate TGA and DSC curves and say about thermal stabilyty because authors plane to use this CD only in water.
- The authors needs to discuss photostability the results more preciously and PH dependency of FL intensity. What surface group responsible for a particular process. It is necessary to compare FTIR before and after to understand these processes. In this case better to include Figure S1A, B to text.
- On Figure 5b it is need to shows the spread of the data. In this interpretation, it is impossible to speak of a linear dependence of the PL intensity (5C) artificially cutting off the data by 18 mg/mL.
Rather, if you show the full range (up to 25 mg/ml), it will be an exponent. -
What concentration of CD in water is used in your experiments. It is necessary to conduct research at various concentrations of CD in solutions. How sensitive your CD will change at different concentrations?
After all, if the concentration of CD is low, then having attached many Cu2+ ions, they will no longer be able to indicate Cu2+ presence. -
What is the possibility of secondary use of your CD materials.
Is it possible to wash them from copper to second using?
Author Response
Reviewer 1
Review 1: Previous works on the microwave pyrolysis process for CD should be included in the introduction section. And the process should be fully explained.
Response 1: We thank the reviewers for their time, literature regarding the excellent attributes of microwave pyrolysis in synthesis of CDs have been included in the introduction section in Line 87 - 92 of the resubmitted manuscript
Review 2: During the microwave pyrolysis process, what are the temperatures that the powder experience? The authors should further elaborate on the microwave pyrolysis process.
Response 2: The formation mechanism of CDs prepared from microwave pyrolysis have not been fully established up to date, judging from previous literatures. The established formation mechanism from hydrothermal-assisted method comprising of decomposition polymerization, carbonization, and carbon dots formation/ growth, thus suffices for the conditions PC undergoes in microwave heating method in formation of CDs as established in : DOI: 10.1039/c9tc01640f; doi/10.1021/acsomega.9b01798; doi.org/10.1021/acsomega.8b03674.
Review 3: The quality (resolution) of TEM image is poor. These images should be replaced with higher-quality images. Furthermore, the TEM images do not support the claim that authors have made regarding the size of the CD. The average size 15 and 42 nm of these CD is not demonstrated. It is need to use Malvern sizer to check size in water solution directly.
Response 3: The suggested improvement of the TEM image has been revised in the submitted manuscript. Furthermore, the average size of 15 and 42 nm have now been properly represented in the revised manuscript. Please note that a Malvern Mastersizer measures particle sizes in the micrometer size range (0.1-3000 μm) and is therefore inappropriate for nanometer range analysis (https://www.montana.edu/eal-lres/instrumentation/malvern.html).
Review 4: It is possible to calculate average size of CD from the XRD data using Seliakov equanation. The authors should also include a histogram of the CD size range.
Response 4: The authors have now calculated the size of PC-CDs using Debye Scherrer equation as presented in the revised manuscript in Line 186 – 187.
Review 5: Experimental parameter such as particle size of the raw material, the density of the pellet, etc., plays an important role in the microwave pyrolysis process. The authors should explain why only a single experimental condition was used to synthesize the CD, and on what basis these experimental conditions were chosen?
Response 5: The scope of manuscript, and literature review of the optimum parameters utilized herein this study will suffice to obtained the required data for the PC-CDs analysis, and sensing attributes. CDs do experience surface deterioration, and graphitization occurs at higher temperature / Microwave power, and longer reaction time, which causes reduction quantum yield as highlighted in these studies: RSC Adv. 7 (2017) 24771–24780; 10.1021/acsomega.7b00551; doi.org/10.1016/j.apsusc.2017.05.036; doi.org/10.1016/j.jpha.2019.02.003.
This work was carried using opened-vessel microwave-assisted synthesis approach, whereby microwave power and time are crucial factors according to these studies: doi.org/10.1016/j.jpha.2019.02.003; doi.org/10.1016/j.microc.2021.106116
Review 6: It is no need to demonstrate TGA and DSC curves and say about thermal stability because authors plane to use this CD only in water.
Response 6: The authors have revised this aspect in the revised manuscript by comparing thermal stability of PC-CDs with other biomass induced CDs.
Review 7: The authors needs to discuss photostability the results more preciously and PH dependency of FL intensity. What surface group responsible for a particular process. It is necessary to compare FTIR before and after to understand these processes. In this case better to include Figure S1A, B to text.
Response 7: The functional groups on PC-CDs responsible for pH dependency on FL intensity are the amino and negatively charge oxygenated groups as described in Line 269 – 272 of the revised manuscript.
Review 8: On Figure 5b it is need to shows the spread of the data. In this interpretation, it is impossible to speak of a linear dependence of the PL intensity (5C) artificially cutting off the data by 18 mg/mL. Rather, if you show the full range (up to 25 mg/ml), it will be an exponent.
Response 8: The authors have corrected this aspect in the revised manuscript, using a non-linear exponential function in the revised manuscript.
Review 9: What concentration of CD in water is used in your experiments. It is necessary to conduct research at various concentrations of CD in solutions. How sensitive your CD will change at different concentrations? After all, if the concentration of CD is low, then having attached many Cu2+ ions, they will no longer be able to indicate Cu2+ presence.
Response 9: The authors have stated the concentration of PC-CDs used in the experiments in Line 151 of the revised manuscript. Moreover, at higher concentration of CDs, they become unstable and tend to aggregate into larger particles size. This further enables direct interaction of more and more functional groups on surface of the CDs as well as the decrease of quantum yield which is ascribed to fluorescence self-quenching : 10.1039/C7RA04781A, doi.org/10.1016/j.jcis.2020.04.004, doi.org/10.1016/j.jcis.2018.03.021, doi.org/10.1016/j.jlumin.2018.02.012. As such, the active sites of CDs will be excessive overpopulated for binding of Cu2+ at higher concentration.
Review 10: What is the possibility of secondary use of your CD materials. Is it possible to wash them from copper to second using?
Response 10: The prepared PC-CDs have huge potential to be washed off the copper ion, through additional sensing activities with few selected organic acids, thus restoring the PC-CDs to its original form for further applications as bio-imaging and patterning agents. However, this scope of work is focused on sensing of copper metal ion and such work will be considered for future studies.
Reviewer 2 Report
The paper focuses on the development of carbon dots (derived from pine cone biomass) with a bimodal size distribution that can be used for the detection of Cu2+. PC-CDs have been characterized by TEM, XRD, Raman, FTIR, TGA, DSC, UV-vis and fluorescence spectroscopy.
The paper falls within the scope of the journal and might attract the interest of its readers. My comments/suggestions are shown below:
- The authors state that “PC-CDs were red shifted with increased emission wavelength from 418 to 430 nm”. However, this is not obvious in Figure 3b. It would appear that this dependency is much more limited compared to similar systems. The authors should comment on this observation.
- How the thermal stability of PC-CDs shown Figure 2b compares with similar systems?
- What tis the origin of the sharp peak in XRD close to 65-70 degrees shown in Figure 1 c?
- Regarding Table 1, biomass generated C-dots with high quantum yield are reported in a number of studies, see for example Nanomaterials 2019, 9, 495. The authors should expand the pool of comparators in table 1.
- C-dots with bimodal size have been reported elsewhere, see for example Green Chemistry 2012, 14, 3141. Is this effect similar to this observed here? The authors should discuss on this trend.
Author Response
Reviewer 2
Review 1: The authors state that “PC-CDs were red shifted with increased emission wavelength from 418 to 430 nm”. However, this is not obvious in Figure 3b. It would appear that this dependency is much more limited compared to similar systems. The authors should comment on this observation.
Response 1: We thank the reviewers for their time, as this aspect have been properly revised, and compared with other studies with significant shift as excitation wavelength increases.
Review 2: How the thermal stability of PC-CDs shown Figure 2b compares with similar systems?
Response 2: The authors have revised this aspect in the revised manuscript, by comparing thermal stability of PC-CDs with other biomass induced CDs.
Review 3: What tis the origin of the sharp peak in XRD close to 65-70 degrees shown in Figure 1 c?
Response 3: The sharp peak in the XRD plot in Fig. 1C emanates from the background noise of the XRD equipment.
Review 4: Regarding Table 1, biomass generated C-dots with high quantum yield are reported in a number of studies, see for example Nanomaterials 2019, 9, 495. The authors should expand the pool of comparators in table 1.
Response 4: The authors have now added more CDs with high quantum yield prepared from biomass, in Table 1of the revised manuscript.
Review 5: C-dots with bimodal size have been reported elsewhere, see for example Green Chemistry 2012, 14, 3141. Is this effect similar to this observed here? The authors should discuss on this trend.
Response 5: The authors have addressed this aspect properly in Line 179, 182 - 184 of the revised manuscript.
Round 2
Reviewer 1 Report
- Authors use Malvern Zetasizer Nano (Malvern, UK) to analyzed Zeta potential of the PC-CDs (line 158-159). Why in answer to my remark number 3 they answer about other Malvern mashine? Malvern Zetasizer Nano can measure particle size from 0.3 nm to 10 μm (https://www.malvernpanalytical.com/en/products/product-range/zetasizer-range/zetasizer-advance-range/zetasizer-lab
Please make measurement of CD size directly by DLS methode using Malvern Zetasizer. It is need to prove by direct measurement that synthesized PC-CDs had a bimodal distribution.
Author Response
Review 1: Authors use Malvern Zetasizer Nano (Malvern, UK) to analyzed Zeta potential of the PC-CDs (line 158-159). Why in answer to my remark number 3 they answer about other Malvern mashine? Malvern Zetasizer Nano can measure particle size from 0.3 nm to 10 μm (https://www.malvernpanalytical.com/en/products/product-range/zetasizer-range/zetasizer-advance-range/zetasizer-lab.
Please make measurement of CD size directly by DLS methode using Malvern Zetasizer. It is need to prove by direct measurement that synthesized PC-CDs had a bimodal distribution.
Response 1: We thank the reviewer for their time and effort in reviewing the revised version of our manuscript. To clarify the previous response: The authors misinterpreted the review based on known analytical instrumentation (Malvern Mastersizer); the original review indicated: “It is need to use Malvern sizer to check size in water solution directly”.
Regarding DLS measurements of the PC-CDs using Malvern Zetasizer, the authors do not have direct access to this equipment for PC-CDs analyses and therefore would need to gain access through alternative means. However, the benefits of these additional analyses would be highly questionable.
Firstly, it should be noted that the DLS technique measures the hydrodynamic diameter as opposed to the particle diameter of particles. The reason for this relates to the measurement technique used in DLS, i.e. the intensity of the scattered light measured. The DLS effectively measures the diffusional coefficients of the particles and subsequently determines hydrodynamic diameters as the diameters of perfect spheres with these diffusional coefficients (https://warwick.ac.uk/fac/cross_fac/sciencecity/programmes/internal/themes/am2/booking/particlesize/intro_to_dls.pdf). These hydrodynamic diameters are equal to or larger than the particles investigated and therefore DLS tend to overestimate particle size distributions.
Secondly, there have been several comparisons of TEM and DLS as particle analysis techniques published – both as technical reports and as journal articles (https://www.lv-em.com/website_lvem/static/pdf/LVEMApplicationNotes/LVEM_Application_Note-Comparison_of_Nanoparticle_Sizing_Techniques_TEM_vs_DLS_vs_AFM.pdf, https://iopscience.iop.org/article/10.1088/1742-6596/733/1/012039, https://pubs.rsc.org/en/content/articlelanding/2014/ay/c4ay01203h, https://www.mdpi.com/1996-1944/13/14/3101, https://www.sciencedirect.com/science/article/abs/pii/S0021979713001756?via%3Dihub); it has been consistently observed that the DLS can at best match the quality of analyses obtained from the TEM. Specifically related to bimodal particle size distributions, it was shown that TEM analysis gives better information regarding particle size and distribution of bimodal nanomaterials in comparison with DLS analysis [https://www.mdpi.com/1996-1944/13/14/3101, https://iopscience.iop.org/article/10.1088/1742-6596/733/1/012039, https://www.sciencedirect.com/science/article/abs/pii/S0021979713001756?via%3Dihub]. The main reason for the reduced ability of DLS to resolve bimodal/multimodal distributions relates to the signal measured during DLS analysis – the intensity. This intensity is extremely dependent on the amount of light scattered by specific particles – the scattering of light is proportional to the sixth power of the particle diameter (https://www.chem.uci.edu/~dmitryf/manuals/Fundamentals/DLS%20measurement%20principles.pdf). This means that a particle that is 10 times larger would have an intensity peak with and area 1,000,000 times larger. This tends to suppress or underestimate smaller particle size mode in multimodal distributions (https://www.mdpi.com/1996-1944/13/14/3101). In the case of our system the 42.1 nm average diameter distribution would have an intensity peak area 451 times greater than the 15.2 nm average distribution, however the largest particles measured (≈ 50 nm) would have a signal peak area at least 15,625 times that of the smallest particle size (≈ 10 nm) intensity peak area measured.
In conclusion the authors believe that including DLS measurements in the final manuscript will not significantly improve the quality of the manuscript and therefore no changes were made to the final manuscript submitted.
Reviewer 2 Report
The authors made a good effort in revising their manuscript
Author Response
The authors would thank the reviewer for their time in reviewing the revised version of the manuscript.
This manuscript is a resubmission of an earlier submission. The following is a list of the peer review reports and author responses from that submission.
Round 1
Reviewer 1 Report
Dear Respected Editor,
In the submission of “One Step Green Synthesis of Water-Soluble Fluorescent Carbon dots and its application in the detection of Cu2+”, Sanni et al., have synthesized water soluble fluorescent carbon dots (CDs) via facile green microwave pyrolysis of pine cone biomass as precursors without any chemical additives, and they characterize it using structural characterization techniques. Finally, they employed this material as a fluorescence probe sensor for detecting Cu2+ ions. Overall, I consider that the content of this paper is proper for the Nanomaterials journal. However, a major revision is required. Please consider the next comments and suggestions:
- Line 46: mM should be corrected to µM
- Fig 1A: TEM and particle size histogram ranges between 1.5 to 4.5 nm?
- The TEM image clearly shows large clusters of around 50 nm (according to the scale bar) plus well-dispersed particles of an Avg.size of 10-15 nm. Authors should revise the size determination, how many particles are included in the determination and what software was used-please includes all these details in the figure captions. I suggest assigning/labeling a couple of particles on the TEM image with scale.
- Fig 1C: I suggest shortening the y-axis, so, the peaks would be clearer.
- “(Figure 3A) displayed a broad band between 280 and 350 nm”. The spectrum broadness in this region is not clear at all. That could be because authors used higher concentration. The UV should be repeated with more representative one.
- Authors claim that the the PC-CD “exhibits bright blue luminescence under UV light irradiation, as depicted in Figure 3A (inset)” The blue light is similar for the whole image not the PC-CD solution?
- Figure S1. (B) and (C). The Y-axis should be corrected to the range of 7000 – 8000
- Sensitivity study: Please represent the Cu2+ concentration in either ppm or µM to make it consistent with the required detection level presented in the introduction. All units throughout the manuscript should be consistent to avoid confusion.
- Schematic illustration showing the mechanistic behavior of C-dots towards copper metal ion.
- Authors cites references 40 and 50 more than 5 times each. Authors should expand their literature search and reduce the redundant citation of the same study many times.